# Visible-Light-Induced Catalytic Selective Halogenation with Photocatalyst

**DOI:** 10.3390/molecules26237380

**Published:** 2021-12-05

**Authors:** Truong Giang Luu, Yongju Jung, Hee-Kwon Kim

**Affiliations:** 1Department of Nuclear Medicine, Molecular Imaging & Therapeutic Medicine Research Center, Jeonbuk National University Medical School and Hospital, Jeonju 54907, Korea; luutruonggiang.hust@gmail.com; 2Research Institute of Clinical Medicine of Jeonbuk National University, Biomedical Research Institute of Jeonbuk National University Hospital, Jeonju 54907, Korea; 3Department of Applied Chemical Engineering, Korea University of Technology and Education, Cheonan 31253, Korea; yjung@koreatech.ac.kr

**Keywords:** halogenation, photoredox catalysis, visible light

## Abstract

Halide moieties are essential structures of compounds in organic chemistry due to their popularity and wide applications in many fields such as natural compounds, agrochemicals, and pharmaceuticals. Thus, many methods have been developed to introduce halides into various organic molecules. Recently, visible-light-driven reactions have emerged as useful methods of organic synthesis. Particularly, halogenation strategies using visible light have significantly improved the reaction efficiency and reduced toxicity, as well as promoted reactions under mild conditions. In this review, we have summarized recent studies in visible-light-mediated halogenation (chlorination, bromination, and iodination) with photocatalysts.

## 1. Introduction

Halogenation is one of the most important modifications in organic synthesis because of its extremely wide applications. Halogen derivatives are useful building blocks in organic synthesis for the construction of complicated, high-activity molecules [1,2,3]. Moreover, as halogenation can be applied to a wide variety of organic compounds without altering their basic structures, halogen-substituted compounds have become popular intermediates for transformation to create different functional groups [4,5,6]. Various areas such as pharmaceuticals, material sciences, industrial chemicals, and bioactive compounds have all benefited from halogen-containing compounds [7,8,9,10]. So far, more than 5000 halogenated natural compounds have been identified, with several of them exhibiting intriguing pharmacological characteristics (Figure 1) [11]. Thus, developing halogenation methods is an interesting area of research, which has received the attention of scientists for decades.

Traditional halogenation methods include addition reactions to multiple bonds, nucleophilic substitution, or radical substitution reactions [12,13,14,15,16,17,18]. One of the most fundamental halogenation reactions in organic chemistry is the addition of halide reagents to C-C multiple bonds. Halogen electrophiles are the most common type of electrophile, and they are commonly employed to generate electrophilic addition reactions to unsaturated carbon [12]. Multiple-bond compounds (alkenes and alkynes) are easily transformed directly to halogenated products by reacting with halogen molecules or hydrohalic acids. However, this technique has significant weaknesses, such as low selectivity, extremely volatility, and the toxic nature of some halogens, and environmental risks [13].

On the other hand, electrophilic substitution and radical substitution reactions are the most feasible and well-recognized approaches for the production of aryl halides, alkyl halides, and many other halide compounds [14]. To produce a carbon-halogen bond, the C-H bond was broken, and then the hydrogen atom was replaced by a halide anion or radical. Halide substitution reactions often require harsh reaction conditions such as high temperature, inert pressure, or an excess of halogen agent and initiator compounds [13,14,15,16,17,18,19,20]. These requirements have increased the purification and treatment costs of the actual halogenation process.

Several classic or modified halogen sources including N-bromo- and N-chlorosuccinimide, plus Selectfluor for halogenation, have also been used. However, the existence of special reagents has limited the application scope and decreased functional group tolerance. Besides, these reactions have been generally carried out under difficult circumstances with poor atom economy [13,14].

Photocatalysis refers to chemical reactions that use light as an energy source. Under irradiation of light, the ground state photocatalyst receives or releases one electron to transfer to the excited state, which subsequently interacts with the substrates or reagents to cause chemical reactions. Generally, photocatalysts can be divided into three main types, including metal complexes, organic dyes, and heterogeneous catalysts [21,22].

In recent years, visible light photocatalysis has emerged as an effective alternative in organic synthesis. Many studies have demonstrated the effectiveness of the visible-light-mediated method in overcoming the inherent disadvantages of traditional organic synthesis methods, such as proceeding under milder reaction conditions, reducing the amount of initiator, introducing outstanding functional group tolerance, and maintaining good regioselectivity [23]. Using photocatalysts for halogenation improves selectivity, allows better reaction control, and lowers costs [24,25].

Many new discoveries in halogenation utilizing visible light via photoredox catalysis have been made in the last decade, and many positive results from reactions with a variety of substrates involving alkyl, aryl, alcohol, carboxyl, etc., have been achieved. In this review, recent advances in halogenation (chlorination, bromination, and iodination) of a variety of organic molecules via photocatalysis are presented.

## 2. Photo-Catalyzed Halogenation of Aliphatic C-H Bonds

Figure 1 shows schematic diagrams of the comparison of the traditional methods with visible-light-induced halogenation of aliphatic C-H bonds.

### 2.1. Chlorination of Aliphatic C-H bBonds

In 2016, Gong Chen and co-workers developed nucleophilic halogenation of tertiary aliphatic C-H bonds [26]. In the reaction, starting substance **1** reacted with LiCl as a chlorinating source in the presence of (Ru(bpy)_3_Cl_2_) **2** as a photocatalyst and azidoiodane in hexafluoroisopropanol (HFIP) under the irradiation of a fluorescent bulb at room temperature to give the corresponding products (Figure 2). This protocol successfully demonstrated site-selectivity for specific tertiary C-H bonds and functional group tolerance. Substrates bearing functional groups such as ester (**3a**, **4a**), ether (**3b**, **4b**), and amide (**3c**–**3e**; **4c**–**4e**) provided the corresponding products with moderate to excellent yields (45–80%). This method was also applied for bromination of tertiary C-H bonds. n-Bu_4_NBr was employed as a brominating source, and the bromination was proved to be more efficient than chlorination and showed better yields.

A plausible mechanism for chlorination is illustrated in Figure 3, in which azidoiodane **5** participated in two processes simultaneously. Under irradiation of visible light and photocatalyst Ru(bpy)_3_Cl_2_
**2**, homolytic break of the I–N_3_ bond of azidoiodane **5** yielded an iodanyl radical **6** and an azido radical. Azidoiodane **5** also reacted with chlorinating source LiCl to generate chloroiodane **7**. Capture of an H atom of substrate **1** by radical **6** gave the intermediate radical **8**. In the meanwhile, chloroiodane **7** provided a Cl atom to radical **8** to form the desired product **3** and then recovered iodanyl radical **6**.

Another photo-mediated C(sp^3^)-H chlorination was reported by Chuo Chen and co-workers in 2017 [27]. In this reaction, aryl ketones such as benzophenone were employed as a photocatalyst to assist in the chlorination of C-H groups in the presence of N-chlorosuccinimide (NCS) as a chloride source under irradiation of a household compact fluorescence lamp (CFL) in acetonitrile at room temperature (Figure 4). The benzylic C-H chlorination was readily performed regardless of the position (ortho, meta, or para) of an electron-withdrawing group on the benzene ring (**12a**–**c**). Chlorinations of the primary and tertiary benzylic C-H groups (**12d**) were successfully achieved, and the ester group at the β-position (**12e**) was tolerated for this protocol. This method was also highly effective for non-benzylic chlorination, when acetophenone was used as the photocatalyst instead of benzophenone (Figure 5). Particularly, chlorination of cyclo-compounds was carried out smoothly at high yield (**15a**–**b**), whereas the dichlorination of the *tert*-butyl group (**15c**–**d**) was conducted with a lower yield.

In 2020, Wu and co-workers reported a novel strategy for benzylic chlorination using *N*-chlorosuccinimide (NCS) as a chloride source and Acr^+^-Mes as a photocatalyst under radiation of blue LED light in dichloromethane (Figure 6) [28]. Several typical alkylbenzene derivatives were tested to assess the scope of this chlorination method. Reaction of toluene **18a** had a higher reaction yield than that of ethylbenzene **18b** (78% for **18a** and 64% for **18b**). Substrates containing different groups such as phenyl **18e** and carbonyl **18d** on the aromatic ring were smoothly converted to target chlorides in moderate to good yields (64–77%), while reaction of nitro group **18c** on the aromatic ring achieved a lower yield (21%).

A plausible mechanism for chlorination proposed by Wu and co-workers is depicted in Figure 7. Visible light excited Acr^+^-Mes **17** to give the charge state Acr**^•^**-Mes**^•+^ 19**, which caused the oxidization of N-chlorosuccinimide (NCS) or substrate **16** to provide Acr**^•^**-Mes radical **20**. This radical **20** reacted with NCS **11** to yield NCS**^•−^ 21** via the SET process and to recover Acr**^+^**-Mes **17**. Then, NCS**^•−^ 21** lost a chlorine anion to give N-centered radical **22,** which underwent the hydro atom transfer (HAT) process with substrate **16** to afford benzylic radical **24**. The radical **24** captured a chloride atom of NCS **11** to form benzylic chloride product **18**.

In the same year, Wei Yu and co-workers developed a method for the chlorination of aliphatic sulfonamides [29]. The chlorination was achieved via a reaction with NaOCl·5H_2_O crystals as a chlorinating agent, NaHSO_4_, and Ru(bpy)_3_Cl_2_ as a photocatalyst under blue LED irradiation at room temperature in a mixture of acetonitrile and water (4:1) (Figure 8). A wide range of sulfonamide substitutes with variations at the sulfonyl moiety were chlorinated at the δ-position with 71% to 95% yields (**27a**–**27e**), while the substituents on the amide moiety led to a significant decrease in reaction yield (**27f**–**27h**).

A probable mechanism of the reaction is presented in Figure 9. Substrate **26** reacted with NaOCl **30** to form *N*-chlorosulfonamides **31**, which was transformed to sulfonamide radical **32** under the effect of the photocatalyst Ru*(bpy)_3_^2+^ **28**, which was generated from Ru(bpy)_3_^2+^ **2** by light. This radical **32** underwent the 1,5-hydrogen atom transfer (1,5-HAT) process to form carbon-centered radical **34** at the C_5_ position. The carbon-centered radical **34** then participated in two reactions. Firstly, radical **34** was oxidized by photocatalyst Ru(bpy)_3_^3+^ **29** to give carbocation **35** and Ru(bpy)_3_^2+^
**2**, and then carbocation **35** obtained Cl anions to form the final product **27**. On the other hand, radical **34** also picked the chloride atom of compound **31** to generate the final compound **27** and sulfonamide radical **32**.

### 2.2. Bromination of Aliphatic C-H Bonds

In 2013, Nishina and co-workers reported mono-bromination of hydrocarbons [30]. In the reaction, starting substances reacted with Br_2_ in the presence of Li_2_MnO_3_ as a photocatalyst under irradiation of fluorescent light under O_2_ pressure to give brominated products (Figure 10). This reaction showed higher selectivity to the secondary C-H bonds of *n*-hexane than to the primary C-H bonds (**37b**), and bromination of the 2-position had priority over that of the 3-position with a ratio of 2:1. Some other compounds, such as adamantane (**37c**), benzine (**37d**), and *tert*-butylbenzene (**37e**), were all tolerated for this method with good to excellent yield (42–93%). Furthermore, *N*-bromosuccinimide, a bromine source, could be used rather than Br_2_ to brominate a wide range of substrates, which broadened the scope and applicability of this method without the need for harsh reaction conditions (**39a**–**b**) (Figure 11).

In 2014, an efficient C-H bond bromination process on aliphatic and benzylic compounds, without the use of an inert environment or anhydrous solvent, was reported by Tan and co-workers [31]. Eosin Y disodium salt, as a photoredox catalyst, and reductive compound morpholine were employed to perform bromination of aliphatic and benzylic compounds under irradiation of an 11 W lamp in a mixture of dichloromethane and water (1:1) at 34 °C for 24 h (Figure 12). Bromination of adamantane derivatives containing ketones, esters, and ether functional groups were successfully achieved (**43a**–**d**) (55–74% yield). The reaction did not occur with unsaturated C-H, but C(sp^3^)-H on toluene derivatives (**43e**–**f**) and alkyls (**43g**–**k**) were brominated with 2–76% yields. Additionally, they applied this bromination method to some useful compounds that could be utilized in the pharmaceutical and medical fields, and a Terpenoid and an Estrone derivative (**43l**) were brominated with good efficiency.

The proposed mechanism of this reaction is shown in Figure 13. Absorbing light of photocatalyst Eosin Y^2−^ **41** formed Eosin Y^2−*^ (singlet) **44**, which underwent an intersystem crossing process (ISC) to generate Eosin Y^2−*^ (triplet) **45**. When CBr_4_ **47** was reduced to CBr_4_^−^ **48** by Eosin Y^2−*^ (triplet) **45**, the C-Br bonds of CBr_4_^−^ **48** became less stable, and a Br atom of CBr_4_^−^ **48** was lost to form CBr_3_ radical **49**. The CBr_3_ radical **49** then captured a proton from substrate **40** to create CHBr_3_ **51** and carbon-radical R**^•^ 50**, which was linked to the free Br atom or received Br from CBr_4_ to give the product RBr **43**. Morpholine **42** reduced Eosin Y^−^ **46** to Eosin Y^2−^ **41** and afforded compound **52**. Compound **52** reacted with morpholine **42** to give radical **53**, which captured a proton from substrate **40** to afford carbon radical R**^•^ 50**.

In 2018, Franzén and co-workers reported a chemoselective protocol for benzylic C(sp^3^)-H bromination without observation of competing arene C(sp^2^)-H bromination [32]. This process was carried out in the presence of NBS as a bromide source, trityl cation (TrBF_4_) as a Lewis acid organocatalyst, and in dichloromethane under irradiation of fluorescent light (55W F. L.) at room temperature (Figure 14). In the reaction of toluene, benzyl bromide was generated in 82% yield (**58a**). The toluene derivatives with different substitutes (including halogen, nitrozo, cyanide, ester, and sulfochloride) were tolerated for this protocol with good to excellent yields (88–91%) (**58b**–**d**). Reaction of ethylbenzene also gave the corresponding product in 92% yield. Naphthalene and heterocycle derivatives were smoothly brominated via this process to produce target compounds in good yield (82–96%) (**58e**–**f**). For the reaction of diphenylmethane (**58g**), the desired benzyl bromide could be observed by ^1^H NMR. However, this bromide was spontaneously hydrolyzed during isolation and purification to produce the corresponding alcohols.

## 3. Photo-Catalyzed Halogenations of Aliphatic Multiple Bonds

Figure 15 shows schematic diagrams of the comparison of the traditional methods with visible-light-induced halogenation of aliphatic multiple bonds.

### 3.1. Chlorination of Aliphatic Multiple Bonds

In 2020, Nicewicz and co-workers developed an organic photoredox catalyst system for the regioselective addition of strong Bronsted acidic nucleophiles such as HCl to alkenes (Figure 16) [33]. Two different techniques were employed for chlorinating β-methylstyrene derivatives using 9-mesityl-10-methylacridinium as a catalyst under irradiation of 450 nm light. In the first method, a reaction with in situ anhydrous HCl (from pivaloyl chloride and 2,2,2-trifluoroethanol (TFE)) in thiophenol and 2,6-lutidine in chloroform was performed under irradiation of 450 nm light. In the second method, substrates reacted with 2,6-lutidine HCl and 4-methoxythiophenol in a mixture of CHCl_3_ and TFE under irradiation of 450 nm light. Reaction using in situ anhydrous hydrogen chloride (HCl) yielded an anti-Markovnikov hydrohalogenation product. For the reaction of styrene substrates with electron-withdrawing groups, the corresponding products were generated with moderate yields (51–99%) (**61a**–**d**), and few to no Markovnikov addition compounds were observed. Chlorination of substrates containing electron-releasing substituents showed lower yields and favored the undesirable Markovnikov reaction (**61e**–**g**). The anti-Markovnikov hydrohalogenation of α-methylstyrene was completed in less than 5 h with a 93% yield using 2,6-lutidine hydrochloride. Reaction of several α-methylstyrene compounds gave the products with more than 60% yields, whereas reaction of mono-substituted styrenyl alkenes provided slightly lower yields.

Vicinal chloro-trifluoromethylation of alkenes was reported by Han and co-workers in 2014 [34]. In the chloro-trifluoromethylation, alkenes reacted with CF_3_SO_2_Cl **64** in the presence of Ru(Phen)_3_Cl_2_
**65** as a photocatalyst and K_2_HPO_4_ as an additive in acetonitrile under visible light at room temperature to give the corresponding products (Figure 17). In general, terminal alkenes showed high reactivity. Alkenes containing *N*-tosyl- and *N*-Boc-protected amines were readily chloro-trifluoromethylated (**66a**–**b**) (99% and 91% yields, respectively), and the reaction of an alkene bearing a phthalimide group generated the corresponding product (**66c**) (88% yield). Notably, unprotected hydroxyl and formyl groups of alkenes (**66d**) were tolerated for the reaction procedure, giving 75% and 83% yields, respectively. Furthermore, reactions of alkenes containing ether (**66e**), ester (**66f**), amide, and halogen functional groups on the aromatic ring generated target compounds in high yields (71–88%).

A proposed mechanism of this reaction is shown in Figure 18. When being exposed to visible light, Ru(Phen)_3_^2+^ **65** became the excited state *Ru(Phen)_3_^2+^ **67**. Reduction of triflyl chloride CF_3_SO_2_Cl **64** by *Ru(Phen)_3_^2+^ **67** was then cleaved to CF_3_**^•^ 69**, SO_2_, and Cl^−^
**70**. After that, *Ru(Phen)_3_^2+^ **67** became the highest oxidation state Ru(Phen)_3_^3+^ **68**. CF_3_**^•^** radical **69** attacked alkene **63** to generate radical intermediate **71**, which was later oxidized by Ru(Phen)_3_^3+^ **68** to give the carbonation intermediate **72** and Ru(Phen)_3_^2+^
**65**. Finally, Cl^−^ anion **70** was captured by carbonation intermediate **72** to produce product **66**.

In 2015, Dolbier and co-workers reported photoinduced atom transfer radical addition (ATRA) reactions of alkenes using fluoroalkylsulfonyl chlorides (CF_3_SO_2_Cl) [35]. For chlorination, alkenes reacted with CF_3_SO_2_Cl in the presence of Cu(dap)_2_Cl **74** as an efficient photocatalyst and K_2_HPO_4_ as a promoter in dichloromethane under irradiation of visible light, which produced the corresponding products in high yields (Figure 19). Various alkenes were successfully tested for the reaction with CF_3_SO_2_Cl to generate target products. Reactions of unsaturated carbonyl substrates such as amides (**75a**–**e**), esters (**75g**–**75h**), and cyanide (**75f**) led to the production of target products in moderate to excellent yields. Unsubstituted and α-substituted substrates smoothly underwent this process, while synthetic yields were considerably decreased to 52%, when the substrate was replaced at the β-position (**75d**). Other fluoroalkylsulfonyl chlorides, such as HCF_2_SO_2_Cl, H_2_CFSO_2_Cl, and CF_3_CH_2_SO_2_Cl, were tested in this reaction process. Even though, it was discovered that their reactions required higher temperatures (108 °C), this reaction procedure of alkenes provided desired products with good to excellent yields (61–98%) (Figure 20). (**77a**–**77i**).

In 2020, Wan and co-workers demonstrated photoredox vicinal dichlorination of alkenes [36]. For this transformation, CuCl_2_ (20 mol%) as a catalyst and hydrochloric acid (2.5 equiv.) as a chlorine source were used for dichlorination in acetonitrile under irradiation of a 38W white LED (Figure 21). A variety of phenolic esters with electron-withdrawing groups (NO_2_, SO_2_, carbonyl, CN, ester, CF_3_, and halides) (**79a**–**c**) and electron-donating groups (phthalimide, *N*-hydroxyphthalimide, acetal, Me, *t*-Bu, and ether) (**79d**–**e**) on benzene rings were well tolerated for this reaction with moderate to good yields (50–71%). Reaction of sulphonamides (**79f**) with free N-H groups was successfully conducted, providing dichlorinated compounds with acceptable yield (75%). The presence of heteroatoms such as oxygen and sulfur had no effect on the efficiency of this reaction, while alkenes with oxidatively labile amine groups were readily converted into dichloride products.

A proposed mechanism of this reaction is shown in Figure 22. When CuCl_2_ **80** was irradiated by visible light, it was excited to CuCl_2_* state **81**. After that, ligand to metal charge transfer (LMCT) excitation occurred, forming chlorine radical **83**, which quickly reacted with alkene **78** to give radical **84**. Finally, radical **84** reacted with CuCl_2_ **80** to afford the desired product **79** and CuCl. Then, oxidation of CuCl by HCl recovered CuCl_2_
**80**.

### 3.2. Bromination of Aliphatic Multiple Bonds

In 2011, Stephenson and co-workers developed a photoredox-catalyzed halogenation via atom transfer radical addition (ATRA) of haloalkanes and α-halocarbonyls to olefins [37]. By using Ir[(dF(CF_3_)ppy)_2_(dtbbpy)]PF_6_ as a photocatalyst and LiBF_4_ as a Lewis acid additive, they carried out the addition of various haloalkanes and α-halocarbonyls to different olefins under irradiation of visible light in a mixture of DMF and H_2_O (1:4) (Figure 23). Using diethyl 2-bromomalonate as a halide source, the reaction of monosubstituted and 1,1-disubstituted olefins was carried out smoothly (67–99% yields). Olefin functional groups that were well tolerated included free alcohols, silyloxy ethers, benzyl ethers, alkyl bromides, esters, enones, carbamates, and aromatic rings (**88a**–**b**). A number of α-halocarbonyls and haloalkanes could be used as halogen sources. A variety of fluorinated compounds were successfully employed for this reaction, generating products with high yields (75–93%) (**88c**–**d**).

A mechanism of this reaction was proposed as shown in Figure 24. Ir^3+^ **87** was changed to excited state Ir^3+^* **89** under irradiation of visible light, and Ir^3+^* **89** subsequently reacted with haloalkane **86** or α-halocarbonyl to give radical **91** and Ir^4+^[X^−^] complex **90**. The electrophilic radical **91** then underwent an atom transfer radical addition (ATRA) process with olefin **85** to generate a new radical **92**. This radical **92** was oxidized by Ir^4+^ and then captured X^−^ to give the product **88**. On other hand, the new radical **92** also received X^−^ from haloalkane **86** or α-halocarbonyl to give the product **88**.

Synthesis of α-bromoketones from olefins was reported by Zhang and co-workers in 2021 [38]. In this method, the reactions of styrenes with CHBr_3_ in the presence of Ru(bpy)_3_Cl_2_ (1.0 mol%) as a photocatalyst and PhI(OAc)_2_ (1.0 equiv) as a promoter in dioxane under irradiation of a blue LED (450–455 nm) was carried out to produce α-bromoketone products in good yields (Figure 25). Using this protocol, various olefin derivatives were transformed to α-bromoketones. Styrene with different substitutes such as methyl groups and halides were readily treated with tribromomethane to give the corresponding products in good to excellent yields (**95a**–**e**) (79–91%). In addition, 2-vinylnaphthalene was transformed to a desired product with high yield (92%) via this protocol (**95f**). This visible-light-irradiation protocol was also applied to the synthesis of α-iodo/chloroketones from olefins, and it successfully provided target products.

A proposed mechanism of this method is illustrated in Figure 26. First, photocatalyst (PC) **2** was activated under irradiation of visible light to produce the excited state (PC)* **28**. The generated (PC)* **28** then reacted with halide reagent **96**, yielding halide radicals (X^•^
**97** and (CHX_2_)^•^ **98**) through C-X bond cleavage. Addition of X^•^ radical **97** to substrate **94** yielded radical intermediate **99**, which was then incorporated with ^3^O_2_ to give intermediate radical **100**. The radical **100** captured a hydrogen atom from (CHX_2_)^•^ radical **98** to form compound **101**, which subsequently underwent a dehydration process to provide the final product **95**.

## 4. Photo-Catalyzed Halogenations of Alcohols

Figure 27 shows schematic diagrams of the comparison of the traditional methods with visible light-induced halogenation of alcohols.

In 2011, Stephenson and co-workers performed halogenation of alcohols in the presence of CBr_4_ or CHI_3_ as halide sources and Ru(bpy)_3_Cl_2_ as a photocatalyst in DMF under radiation of blue LED irradiation at room temperature (Figure 28) [39]. Substrates bearing various functional groups such as ethers, silyl ethers, alkene, alkynes, carbamates, and phenols were tolerated for this reaction procedure (**107a**–**h**). In this reaction, primary alcohols were successfully converted to the corresponding halides with yields ranging from 77 to 98%. Reactions of secondary alcohols were smoothly conducted for the bromination and iodination processes (**107i**–**j**), although the reaction rates were slower than those of primary alcohols.

A possible mechanism was proposed as shown in Figure 29. Under visible light irradiation, the photocatalyst Ru(bpy)_3_^2+^ **2** was changed to excited state Ru(bpy)_3_^2+*^ **28**, which underwent single-electron oxidation by CBr_4_ **47** to generate Ru(bpy)_3_^3+^ **29** and electron-deficient radical ^•^CBr_3_
**49**. The ^•^CBr_3_ radical **49** then reacted with DMF **108**, resulting in stable radical **109**. Ru(bpy)_3_^3+^ **29** was reduced by radical **109** to return Ru(bpy)_3_^2+^ **2** and produced intermediate **110**. On the other hand, intermediate **110** was also generated through the reaction of radical **109** with CBr_4_ **47**. At this point, there are two possible ways to afford the target product. The first way involved the reaction of alcohol with compound **110** to create intermediate **112**. In the second process, the bromide anion directly attacked intermediate **110** to generate Vilsmeier–Haack reagent **111**, which then reacted with alcohol **106** to form intermediate **112**. Finally, the SN_2_ substitution reaction of **112** with bromide anion provided the desired product **107**.

Bromination of alcohols using metal-free organic photocatalyst was demonstrated by Li and co-workers in 2019 [40]. The bromination reaction of alcohols was carried out in the presence of CBr_4_ as a bromide source and 4,7-diphenyl-2,1,3-benzothiadiazole (Ph-BT-Ph) as a photocatalyst under blue LEDs irradiation in DMF at room temperature to yield the corresponding products (Figure 30). Both primary and secondary alcohols were readily converted into desired bromides in the use of Ph-BT-Ph. Reaction yields of primary alcohols (**116a**–**c**) were somewhat greater than those of secondary alcohols (**116d**–**e**). It was noted that formate ester was observed as a minor side product from the reaction of alcohols, and, in the reaction of cyclododecanol, cyclododecyl formate (**116f**) was generated as a main product. No photobleaching impact of photocatalyst was discovered.

A mechanism was proposed as shown in Figure 31. Under irradiation of visible light, one electron was transferred from the lowest unoccupied molecular orbital (LUMO) of the photocatalyst Ph-BT-Ph **115** to CBr_4_
**47** to afford ^•^CBr_3_ radical **49** and Br^−^. DMF captured radical **49** to give intermediate **109**, which subsequently delivered an electron, resulting in iminium compound **110**. The bromide ion reacted with intermediate **110** to generate Vilsmeier–Haack reagent **118**, which then interacted with alcohol **114** to produce the desired compound **119**. In another pathway, reduction of CBr_4_ **47** by photocatalyst Ph-BT-Ph **115** gave carbene CBr_2_ **120**. Then, reaction of carbene CBr_2_ **120** with DMF produced CO and (dibromomethyl) dimethylamine intermediate **121**, which was also converted to Vilsmeier–Haack reagent **118** after losing one bromide atom.

## 5. Photo-Catalyzed Halogenations of Carboxylic Acids

Figure 32 shows schematic diagrams of the comparison of the traditional methods with visible-light-induced halogenation of carboxylic acids.

### 5.1. Chlorination of Carboxylic Acids

In 2016, Glorius and co-workers reported photocatalytic Hunsdiecker-type decarboxylative halogenation (bromination, chlorination and iodination) of alkyl carboxylic acids [41]. Diethyl bromomalonate, *N*-chlorosuccinimide (NCS), and *N*-iodosuccinimide (NIS) were used as halide sources, Cs_2_CO_3_ was a promoter, and [Ir(dF(CF_3_)ppy)_2_(dtbbpy)]PF_6_ was a photocatalyst in chlorobenzene to perform this decarboxylative halogenation of carboxylic acids under irradiation of blue LEDs (lmax = 455nm) (Figure 33). Primary, secondary, and tertiary carboxylic acid substrates were all tolerated for this method. A broad range of functional groups such as esters, protected amines, aryl, and silyl ethers were successfully used in this protocol to produce target products with good to high yields (**124a**–**w**) (25–86%). It was discovered that the reaction could also be achieved with excellent product yields in ethyl acetate instead of chlorinated solvents. Besides, this reaction could be conducted on a gram scale (5 mmol) without reducing the yield, even though a longer reaction time was required.

A proposed mechanism for this method is illustrated in Figure 34. Photocatalyst Ir^III^ **87** was transformed to Ir^III^* **89** under the irradiation of visible light. Photoexcitation of photocatalyst Ir^III^* **89** facilitated decarboxylation of substrate **122** to give Ir^II^ **125** and appropriate alkyl radical **126**, which captured a halide atom from the halide source **123** to give the final product **124** and malonyl radical **127** as a byproduct. The malonyl radical **127** received one electron from Ir^II^ **125** to recover photocatalyst Ir^III^
**87** and yielded the malonate anion **128**.

### 5.2. Bromination of Carboxylic Acids

Another decarboxylative bromination of carboxylic acids using potassium bromide was reported by Uchiyama and co-workers in 2020 [42]. In this protocol, a series of sterically hindered primary, secondary, and tertiary carboxylic acids, bearing different structures such as acyclic, cyclic, caged, and bridgehead, were treated with potassium bromide in the presence of (diacetoxyiodo)benzene as a photocatalyst in CH_2_Cl_2_ or PhCF_3_ under irradiation of a ceiling fluorescent light at room temperature to generate the corresponding products without rearrangement or fragmentation (Figure 35). Substrates bearing nitro, ester/lactone (**131d**–**e**), amide/sulfonamide/2-nitrophenylsulfonyl (nosyl)/imide (**131i**), ketone (**131b**, **131f**–**g**), and bromide/fluoride (**131h**) functionalities were tolerated for this brominating procedure. Additionally, reaction of carboxylic acid with the extremely radical-sensitive ether group was also successfully achieved with 87% yield.

A possible mechanism was proposed as shown in Figure 36. The hypervalent iodine reagent PhI(OAc)_2_ **130** reacted with substrate **129**, followed by treatment with KBr, to provide intermediate **133**. Intermediate **133** was triggered by visible light to yield ^•^Br radical and iodo-radical intermediate **134**. The I-O bond in radical **134** was cleaved to afford acyloxy radical **135**. Removal of CO_2_ from **135** gave cyclopropyl radical **136**. At last, radical **136** captured the bromide source to generate the desired product **131**.

## 6. Photo-Catalyzed Halogenations of Aromatic C-H Bonds

Figure 37 shows schematic diagrams of the comparison of the traditional methods with visible-light-induced halogenation of aromatic C-H bonds.

### 6.1. Halogenation of Aromatic C-H Bonds

In 2015, Ghosh and co-workers reported halogenation of aromatic C-H bonds utilizing Cu-MnO as a heterogeneous catalyst [43]. In this methodology, Cu-MnO as a catalyst, *N*-halogen succinimide as a halide source, and O_2_ as an oxidant reagent were employed to perform the halogenation in nitrobenzene at 125 °C under irradiation of visible light to provide the corresponding products (Figure 38). It was reported that good yield and high regioselective halogenation of aromatic C-H bonds can be well achieved with other transition metal catalysts (Pd, Au, Ru, Co) [44,45,46,47]. However, in this study, a heterogeneous Cu-MnO catalyst was employed due to the cost-efficiency, ubiquity, and versatility properties of Cu. This halogenation (chlorination, bromination, and iodination) produced monohalogenated products selectively in moderate to high yields (32–83%). In this method, chlorination performance was generally better than that of bromination and ionization. Using this protocol, the monohalogenated products from substrates bearing electron-donating groups such as methyl and methoxy groups in the para position of the aromatic ring were prepared with high yield (**136d**–**i**), whereas the monohalogenated products from substrates bearing an electron-withdrawing group, such as the trifluoro-methyl group (**139j**–**l**), were obtained with moderate to good yields (32–44%).

A plausible mechanism of this halogenation as proposed by Ghosh and co-workers is shown in Figure 39. After oxidization of Cu^I^
**140** to Cu^II^
**141** by O_2_, Cu^II^ **141** reacted with 2-phenylpyridine **137** and halide ion, which was generated from NXS **138** under irradiation of visible light to produce complex **142.** A single electron transfer (SET) process between the phenyl ring and Cu^II^ caused complex **142** to become cationic radical **143**. Then, intramolecular transfer of the halide anion to the phenyl ring of **143** yielded compound **144** and recovered Cu^I^ **140**. Finally, **144** underwent another single electron transfer (SET) process to lose a proton to give the desired product **139**.

In 2017, Wu and co-workers demonstrated halogenation of quinoline using a photoredox process in mild conditions [48]. This halogenation was achieved in the presence of alizarin red S as a photocatalyst, FeCl_3_ as a catalyst, K_2_S_2_O_8_ as an oxidant, and potassium halides as halogen sources under irradiation of CFL in water at room temperature (Figure 40). Being abundant, readily obtainable, inexpensive, and non-toxic, water is more environmentally friendly compared to other organic solvents. However, most organic substrates are poorly soluble in neat water, and, thus, the reactions that take place in neat water are often inefficient and generate the products with low yields. On the other hand, organic substrates are easily soluble in organic solvents such as DMF, and organic solvents help increase the reaction yields even though they are harmful and not environmentally friendly. Wu’s method used water as a solvent, but could overcome the disadvantage of aqueous solvents to still achieve high yields. Using this reaction procedure, all target compounds were readily prepared in good to outstanding yields, but the effect of substitute groups on the benzene ring of benzamides on the reaction efficiency was not clearly understood. Bromination was conducted more effectively than iodination. Substrates with methoxy, methyl, and halide groups on the quinoline ring yielded the desired products in good yields (75–98%) (**147a**–**f**). Additionally, the reaction of heterocyclic amide substrate produced the halogenated product in high yields (70–83%) (**147g**–**j**).

A plausible mechanism is illustrated in Figure 41. Initially, the household light excited photocatalyst alizarin red S **146**, resulting in the excited state alizarin red S* **148**. Alizarin red S* **148** was reductively quenched by Br^-^, affording a ^•^Br radical **151** and an alizarin red S^*−^
**149**. K_2_S_2_O_8_ **152** then oxidized alizarin red S**^•^**^−^
**149** to recover the ground state alizarin red S **146** and provided SO_4_**^•^**^−^
**153**. FeCl_3_ incorporated with substrate **145** to give chelated compound **150**, which reacted with ^•^Br radical to give intermediate radical **154**. Then, intermediate radical **154** was oxidized by SO_4_**^•^**^−^ **153** to generate cation intermediate **155**. Finally, intermediate **155** interacted with a chloride anion to give the final product **147**, while recovering FeCl_3_
**156**.

Combining photocatalysis and biocatalysis for halogenation of aromatic compounds was described by Gulder and co-workers in 2018 [49]. They employed vanadium-dependent haloperoxidase (VHPO) from Acaryochloris marina (*Am*VHPO) and Curvularia inaequalis (*Ci*VHPO) as biocatalysts, flavin mononucleotide (FMN) as a photocatalyst, and KBr or KCl as halide providers for halogenation under irradiation of blue LEDs in a mixture of MES buffer (pH = 6.0) and MeCN to yield the corresponding products (Figure 42). The method was highly effective for the bromination process. Reactions of substrates with benzene ring containing methoxy substituents provided target products in excellent yield (99%) (**158a**–**b**). Substrates with a heterocycle ring were halogenated with moderate performance (**158d**). Regarding chlorination, this method was less efficient for one- and two-substituent derivatives (**159b**–**c**).

In 2018, Ghosh and co-workers reported extensive application of Cu-MnO catalyst for halogenations of anilides and quinolones [50]. They used Cu-MnO as a catalyst and *N*-halosuccinimide as a halogenating source in acetonitrile under visible light irradiation to achieve halogenations of anilides and quinolines with good regioselectivity (Figure 43 and Figure 44). For anilide derivatives, reaction of para-substituted substrates containing both electron-withdrawing and electron-donating groups, such as anilides bearing isopropyl, *tert*-butyl, hexyl, and cyclohexyl groups in the amide chain and chloro, bromo, fluoro, and trifluoromethyl groups in the phenyl ring in acetanilide, successfully produced mono ortho-halogenated products in high yields (81–98%) (**161a**–**h**). The protocol showed that halogenation of 8-aminoquinoline amides with a variety of functional groups was also successful and worked well with an aryl amide group and an alkyl amide group (**163a**–**i**). Benzamides containing both electron-donating and electron-withdrawing groups were readily halogenated. In addition, the reactions of alkyl amides such as acetamide, cyclopentanecarboxamide, and decanamide were successfully conducted to give desired products.

A possible mechanism for this halogenation was proposed as shown in Figure 45. For anilides, nitrogen of substrate **160** was coordinated with Cu^II^ to generate complex **164**, and then visible light caused the N–Cu bond of **164** to homolytically break to give the nitrogen radical **165** and Cu^I^, which was oxidized by O_2_ to recover Cu^II^. The radical **165** was subsequently converted to intermediate aryl cation radical **166**. Reaction of **166** with the halide radical produced from *N*-halosuccinimide **138** under visible light occurred, followed by rearomatization to afford the final product **161**. For the 8-amidoquinoline derivatives, a similar chemical mechanism was presented. This method showed regioselective addition of halide radical to the amide group at ortho- and para-positions, the most electrophilic locations.

### 6.2. Chlorination of Aromatic C-H Bonds

In 2016, Konig and co-workers proposed chlorination of arenes via reaction with *N*-chlorosuccinimide (NCS) or *N*-chloramines in the presence of [Ru(bpy)_3_]Cl_2_ as a photocatalyst and ammonium peroxodisulfate as an oxidant under irradiation of blue LEDs in a mixture of acetonitrile and water (4:1) (Figure 46) [51]. In these reaction conditions, substrates with electron-donating groups such as anisole (**174a**–**b**), methoxybenzene, phenol, and acetanilide were chlorinated in good yields (92–95%) via treatment of both *N*-chlorosuccinimide **11** and *N*-chloramines **173**. However, reaction of aromatic amines (**174c**), xylene, and toluene (**174e**) provided chlorinated products with low yields when *N*-chloramines were used. Besides, electron-poor substrates such as chlorobenzene (**174k**) did not provide chlorination. When compared to *N*-chloramines **173**, the electron density on the nitrogen atom was significantly lowered by two electron withdrawing groups in *N*-chlorosuccinimide **11**, and it was able to chlorinate fewer electron-dense substrates such as xylene and toluene by NCS.

Selective chlorination of aryl C-H bonds using NaCl as a chlorine source, Ru(bpy)_3_Cl_2_∙6H_2_O as a photocatalyst, and Na_2_S_2_O_8_ as an oxidant under irradiation of a blue LED in a mixture of acetonitrile and water (1:1) at room temperature and air pressure was reported by Hu and co-workers in 2017 (Figure 47) [52]. In this reaction, substrates bearing electron-donating groups such as isopropyl (**176a**–**b**) and methoxy groups (**176c**–**d**) on the phenyl ring were chlorinated in good yields (82–94%). Both para- and ortho-chlorinated products were also readily formed. Chlorination of substrates bearing -CN, an ester, or a halogen connected to the benzene ring via an alkyl chain worked well in good yields (89–91%) (**176e**–**h**). On the other hand, the substrates containing only an electron-withdrawing group such as nitrobenzene and (trifluoromethoxy)benzene were not tolerated for this method. However, the yields from these substrates were improved when combining an electron-withdrawing group and an election-donating group on the aryl ring.

A proposed mechanism of this reaction was presented in Figure 48. Under light, the photocatalyst Ru^II^ **2** was excited to *Ru^II^
**28**, which reacted with Na_2_S_2_O_8_ **152** to generate Ru^III^ **29** and SO_4_**^•^**^−^. SO_4_**^•^**^−^ directly oxidized Ru^II^ **29** to give Ru^III^
**2**. Then, Ru^III^ **29** reacted with Cl^−^ to regenerate Ru^II^ **2** and form Cl^+^ **177**. Cl^+^ **177** then reacted with aromatic compounds **175** to give the chlorination products **176** through the electrophilic addition of **178**.

Lamar and his co-workers developed chlorination of arenes and heteroarenes using organic dyes as visible light photoredox catalysts in 2019 [53]. The reaction was carried out via treatment with NCS as a chlorine source in the presence of methylene green as an organic dye photocatalyst under irradiation of a white LED in acetonitrile (Figure 49). Under the optimized conditions, reactions of disubstituted benzene derivatives bearing activating (electron-donating) and deactivating (electron-withdrawing) groups readily provided the corresponding products with good to high yields (58–86%) (**181b**–**d**). Various heteroarenes such as pyridine, pyrrole, indazole, and indole were tolerated for the reaction (**181e**–**h**), giving chlorinated products with moderate to excellent yields (63–93%).

A plausible reaction mechanism was proposed as shown in Figure 50. Irradiation of light allowed methylene green to transfer from ground state methylene green **180** to excited state methylene green **182**. Then, **182** led to the single-electron oxidation of NCS **11**, giving cationic radical **184** and providing reduced state methylene green **183**. The radical **184** reacted with substrate **179** to generate arene chloride cation intermediate **186** and charged succinimide **185**. Capture of a proton from intermediate **186** by anion **187** yielded final product **181**. In addition, the reduced state methylene green **183** was oxidized to give back ground state methylene green **180** and to provide succinimide anion **187** by other oxidants (for example ammonium peroxodisulfate or oxygen gas in air) or by charged succinimide **185**.

Hammond and co-workers proposed a novel strategy for chlorination of arenes and heteroarenes using brilliant green (BG) in 2019 [54]. The chlorination was conducted via treatment of trichloroisocyanuric acid (TCCA) as a chlorine source in the presence of brilliant green (BG) under irradiation of white LEDs in acetonitrile (Figure 51). For monosubstituted benzene, reactions of substrates with an electron-donating group smoothly afforded the corresponding products with high yields (74–92%) (**192a**–**b**), while the chlorination of substrates with an electron-withdrawing group such as nitrobenzene (**192c**) and acetophenone (**192d**) was unsuccessful. On the other hand, reactions of naphthalene derivatives bearing electron-withdrawing groups or electron-donating groups successfully yielded monochlorinated products in good to high yields (63–96%) (**192e**–**f**). Various functional groups including halogens, carbonyls (ketone, aldehyde and amide), phenol, ethers, amines, nitro, nitrile, and benzylic C-Hs were tolerated for this method. Additionally, heterocyclic compounds were successfully chlorinated using this reaction as compared with the failure of the general reaction of TCCA using acidic conditions.

A possible mechanism was proposed as shown in Figure 52. Photocatalyst brilliant green BG **190** was transferred to excited state BG* **193** by visible light, and reaction of excited state BG* **193** with TCCA **191** caused the single-electron oxidation to generate electrophilic chlorine species **195** and reduced state photocatalyst BG^−^
**194**. **195** reacted with substrate **189** to give chlorination product **192** via electrophilic aromatic chlorination, and to provide **196**, which reacted with TCCA **191** to give **197**. The reduced state photocatalyst BG^−^ **194** was oxidized by **196** or O_2_ to return to its ground state BG **190**.

In 2020, Lamar and co-workers reported FDA-certified food dye mediated-chlorination of aromatic and heteroaromatic substrates [55]. The chlorination of aromatic compound was achieved via reaction with *N*-chlorosaccharin (NCS) as a chloride source in the presence of Fast Green FCF as photoredox catalyst under irradiation of white LED in acetonitrile (Figure 53). Reaction of substrate with at least one electron-withdrawing group produced chlorinated product in moderate to good yields (47–70%). Electron-rich aromatics (**201a**) and naphthalene derivatives (**201f**–**g**) were readily chlorinated to give monochlorinated products in high to exceptional yields (90–94%). Pyrrole (**201e**), indole, indazole, and pyridine heteroaromatic substrates were also well tolerated for this reaction procedure, which showed considerably higher efficiency than uncatalyzed reaction processes. A reaction using 1,3-dichloro-5,5-dimethylhydantoin (DCDMH)/Brilliant Blue FCF system showed similar results to those of the NCS/Fast Green FCF system. However, a study of dichlorination reactions indicated that the chlorination using a DCDMH/Brilliant Blue FCF system was more efficient than chlorination using an NCS/Fast Green FCF system.

A plausible mechanism is shown in Figure 54. Chlorination of brilliant blue **202** by DCDMH **203** was carried out to generate intermediate **204**, which was subsequently converted to dichlorinated sulfonphthalein **205**. Compound **205** then provided the electrophilic chlorine to aromatic compound **199** or heterocycle arenes to give chlorinated product **201**, and, after that, **205** became a monochlorinated sulfonphthalein species **206**, which obtained a chlorine atom from DCDMH **203** to recover dichlorinated compound **205**.

### 6.3. Bromination of Aromatic C-H Bonds

In 2011, Fukuzumi and co-workers developed selective bromination of aromatic hydrocarbons and thiophenes via reaction with a solution of 50% of HBr in O_2_-saturated acetonitrile in the presence of [Acr^+^-Mes][ClO_4_^−^] as a photocatalyst under irradiation of a xenon light (Figure 55) [56]. For methoxy-substituted benzenes, bromination reactions were successfully achieved regardless of position or number of substituents. The reaction was highly selective, and the bromination yield was more than 99% (**208a**–**b**) without observation of dibromo- or tribromo-derivatives during the process. Toluene derivatives were also brominated (**208c**), even though the yield of bromination in the presence of methyl-substituted groups was lower than that of methoxy-substituted benzenes (**208d**). Besides, brominations of thiophenes were readily achieved with high yield (81–99%).

A plausible mechanism of this bromination of aromatic hydrocarbons is presented in Figure 56. Under irradiation of light, intramolecular electron transfer caused Acr^+^-Mes **17** to become the excited state Acr**^•^**-Mes**^•+^ 19**, which oxidized substrate TMB **207** to give radical cation **209**. At the same time, Acr**^•^**-Mes**^•+^ 19** reduced O_2_ to provide radical HO_2_**^•^ 212**. Reaction of **209** with Br^−^ generated the aromatic ring radical **210**, and then radical **210** reacted with HO_2_**^•^ 212** via dehydrogenated process to give brominated product **208** and H_2_O_2_. Besides, when H_2_O_2_ interacted with HBr and substrate **207**, another brominated compound **208** and H_2_O were obtained.

An efficient strategy for bromination of phenols was presented by Xia and co-workers in 2014 [57]. The reaction was carried out with CBr_4_ in the presence of Ru(bpy)_3_Cl_2_ (5.0 mol%) under visible light irradiation (blue LEDs, λ_max_ = 435 nm) in acetonitrile (Figure 57). Both electron-withdrawing and electron-donating groups as substituents in the benzene ring were tested. Reactions of aromatic substrates bearing TMS (trimethylsilyl), TBS (*tert*-butyldimethylsilyl), MOM (methoxymethyl), and THP (tetrahydropyranyl) groups (**215a**–**e**) at the para- and ortho-positions yielded 2- and 4-bromophenol in good to outstanding yields (58–97%), respectively. TMS and methyl-protected naphthalen-2-ol were readily employed to produce 1-bromonaphthalen-2-ol and 1-bromo-2-methoxynaphthalene in high yields (76–98%), with great selectivity (**215f**–**g**). Additionally, reactions of Bn or Ms-protected phenols gave target 2- and 4-bromophenol derivatives with no loss of Bn or Ms groups in good yields (**215h**–**i**).

A mechanism of this conversion was shown in Figure 58. First, visible light excited Ru(bpy)_3_^2+^ **2** to give Ru(bpy)_3_^2+*^
**28**. Oxidation reaction of Ru(bpy)_3_^2+*^ **28** with CBr_4_ occurred as oxidative quenchers generated Br^−^, ^•^CBr_3_ and Ru(bpy)_3_^3+^
**29**. Then, anion Br^−^ was oxidized by Ru(bpy)_3_^3+^ **29** to produce Br_2_ in situ for bromination of phenols and alkenes, and to regenerate Ru(bpy)_3_^2+^ **2**.

Employment of microporous organic polymers (MOPs) for selective bromination of aromatic compounds was reported by Zhang and co-workers in 2016 [58]. The bromination was achieved via reaction with HBr as a bromine source in the presence of MOPs as heterogeneous photocatalysts and molecular oxygen as a clean oxidant under irradiation of visible light in acetonitrile (Figure 59). Electron-rich aromatic compounds were readily brominated in good to excellent yields (55–89%). In addition, benzene, naphthalene, thiophene, and 3-methylbenzo[b]thiophene derivatives were well tolerated for this protocol (**221a**–**e**). In this study, it was discovered that the methyl group on the aromatic ring led to lower bromination efficiency than that of methoxy groups. Toluene was not brominated under the same reaction conditions (**221f**).

A proposed mechanism of this reaction is shown in Figure 60. Under irradiation of light, MOPs material **220** oxidized substrate (TMB) **219** to generate cationic radical TMB^•+^ **222**. Reaction of TMB^•+^ **222** with Br**^−^** anion from HBr formed TMB^•^-Br radical **226**. Besides, the activation of oxygen by MOPs material **220** gave its active forms of O_2_^•**−**^ and ^1^O_2_. These activated oxygen species oxidized TMB^•^-Br radical **226** to create the desired product **221**, along with H_2_O_2_ **227** as a byproduct. However, H_2_O_2_ **227** also reacted with substrate TMB **219** and HBr in a minor side reaction to form final product **221**.

Organic dye-catalyzed bromination of arenes and heteroarenes was demonstrated by Lamar and co-workers in 2018 [59]. In this method, arenes reacted with *N*-bromosuccinimide (NBS) in the presence of erythrosine B as a photocatalyst and ammonium peroxodisulfate as an oxidant reagent under irradiation of white LED light in acetonitrile to give the corresponding products (Figure 61). Erythrosine B is a xanthene dye, commonly used in daily life as a food colorant and a painting ink, and it is difficult to degrade under visible light without other supporting agents. In this study, the optimization reactions using Erythrosine B were performed in 2, 6, and 24 h, which gave the target products in high yields. Based on these results, it can be believed that it is stable under irradiation of light around 24 h. Under the optimized reaction conditions, naphthalene and anisole derivatives were efficiently brominated on the aromatic ring without byproducts. Brominations of aryl ether-containing substrates, phenol derivatives, aniline derivatives, acetanilide, and the anesthetic lidocaine were also successfully conducted. However, arenes with electron-withdrawing groups such as chlorobenzene and nitrobenzene did not work well. On the other hand, reactions of N-containing heteroarenes (pyrrole, pyrazole, indole, and indazole) successfully afforded brominated products in acceptable to good yields (37–99%).

They proposed a possible reaction mechanism as shown in Figure 62. Under irradiation of light, ground state erythrosine B **229** was converted to excited state erythrosine B **231**, which oxidized the nitrogen of *N*-bromosuccinimide **38** to provide cationic radical **233** and reduced state erythrosine B **232**. The cationic radical **233** reacted directly with arene substrate **228** to generate the desired product **230** and charged succinimide species **234** as a byproduct via electrophilic aromatic bromination. The succinimide **234** or external oxidant such as O_2_, (NH_4_)_2_S_2_O_8_, or H_2_O_2_ oxidized reduced state erythrosine B **232** to return to its ground state erythrosine B **229** and generate succinimide anion **236**.

Anthraquinones were employed as photocatalysts in bromination of arenes and heteroarenes by König and co-workers in 2018 [60]. The bromination reaction was carried out by using sodium bromide as a bromide source, sodium anthraquinone sulfonate (SAS) as a photocatalyst, and TFA acid as an activator for anthraquinone under irradiation of LEDs in a mixture of acetonitrile and water (1:1) (Figure 63). All the methoxy arenes were successfully brominated in high isolated yields (56–100%) (**239a**–**c**). Bromination of substrates bearing Boc-protected amine (**239e**) was readily achieved in good yield (73% yield), even though there was an acidic environment. In contrast, reactions of substrates with an electron-withdrawing substituent provided desired products with lower yields. In addition, heteroarenes such as indole derivative **239g** and benzimidazole **239h** derivatives were also brominated in modest yields (49–63%). This procedure using SAS was also successful to oxidize pyrazole derivatives to generate the corresponding products in high yields. Besides, several bioactive compounds such as phenazone, tramadol, and alkaloid strychnine were tested in this procedure and yielded the brominated products in good to excellent yields (25–97%).

A possible mechanism of this bromination reaction was proposed as shown in Figure 64. TFA **245** reacted with SAS **238** to give the active state SAS-H^+^ **240**, which was stimulated by visible light to yield the triplet state SAS-H^+*^ **241**. SAS-H^+*^ **241** caused oxidation of arenes **237** to produce radical cation **244**, which in turn reacted with bromide anions and HO_2_^•^ **243** to afford the brominated product **239** and hydrogen peroxide. Besides, SAS-H^•^ **242** radical was oxidized by O_2_ to regenerate SAS **238**.

Lei and co-workers described selective bromination at the C_5_ position of 8-aminoquinoline amides in 2019 [61]. In the reaction, aminoquinoline amides reacted with CBr_4_ as a bromine source in the presence of 10-phenylphenothiazine (PTH) as an organophotoredox catalysis and K_2_CO_3_ as the base under the irradiation of blue light in acetonitrile to give the corresponding products (Figure 65). Both linear and cyclo alkyl-substituted carboxamides were successfully brominated with good or moderate yields (56–83%). Reaction of dodecyl and bulky group-substituted carboxamides produced desired products in 88% and 83% yields, respectively (**249a**–**b**). However, this process was less effective for aryl-substituted carboxamide substrates. Target product was obtained in 80% yield by reaction of substrate with phenyl (**249c**), while brominated product was prepared in only 16% yield when the phenyl group was replaced by para methoxy groups (**249d**).

A plausible mechanism of this bromination reaction proposed by Lei and co-workers is shown in Figure 66. Visible light excited photocatalyst PTH **248** to PTH* **250**, which was then oxidatively quenched by CBr_4_ to provide ^•^CBr_3_ radical **252** and bromide anion. ^•^CBr_3_ radical **252** reacted with substrate **247** to give radical intermediate **253**, which underwent single electron transfer with PTH^*+^ **251** to form radical intermediate **254** and recover ground state photocatalyst PTH **248**. Reaction of radical **254** with Br^−^ anion produced compound **255**, which was then used in the hydro atom transfer (HAT) process to prepare the final product **249**.

Bromotrichloromethane (BrCCl_3_)-mediated efficient and regioselective mono-bromination of electron-rich arenes was developed by Loh and co-workers in 2021 [62]. Reaction of arenes and heteroarene with BrCCl_3_ as a bromine source in the presence of Ru^II^ bipyridyl complex photocatalyst and 2-bromopyridine under irradiation of white light (23W) and air atmosphere gave the corresponding products (Figure 67). Various electron-rich arenes and heteroarenes bearing different substitute groups, such as amino, *N,N*′-dimethyl, hydroxyl, methoxy, and other heterocyclic amino groups, were successfully brominated (**257a**–**j**). Brominations of protected anilines (**257f**) with acid-sensitive protective groups (BOC) and oxidation-sensitive groups (hydroxyl) of the phenol ring (**257j**) were also readily modified to provide target products. However, reactions of substrates baring various functional groups, such as naphthalene, weak electron-withdrawing groups (EDGs), or the combination of EDG and electron-withdrawing groups, did not work well to give desired products.

They proposed a probable mechanism for this protocol as described in Figure 68. Light irradiation made photocatalyst Ru^II^
**2** excited to Ru^II*^ **28**. The substrate **256** was oxidized by Ru^II*^
**28** to give radical cation **259** and Ru^I^
**258**, which was then oxidized by O_2_ to Ru^II^ **2**. The radical cation **259** was converted to cation radical **260**, which in turn reacted with BrCCl_3_ **261** to yield intermediate **264**. Intermediate **264** was deprotonated by a base to afford the brominated product **257**.

### 6.4. Iodination of Aromatic C-H Bonds

In 2019, König and co-workers demonstrated oxidative iodination of arenes [63]. The iodination was achieved via treatment of arenes with I_2_ as an iodine source, TFA, O_2_ as an oxidizing agent, and anthraquinone (AQ) as a photocatalyst under irradiation of LED (400 nm) in benzene (Figure 69). In the reaction, synthetic protocol allowed arenes-bearing electron-donating groups to readily produce the desired iodinated products in good to high yields (56–96%). When arenes9bearing methyl groups were used, they were successfully iodinated without generation of any side reactions on the benzylic position. Besides, reaction of both trimethylbenzene and tert-butyl substituted arenes also worked well, although they have potential steric effects. It was found out that the acid-sensitive ester functionality (**267d**) was unchangeable when conducting this reaction method. In addition, nitrogen-containing compounds in the substrate (**267e**) were also stable during the reaction, which successfully afforded target products in high yields (86–92%).

A plausible mechanism of this iodination was presented in Figure 70. Reaction of photocatalyst AQ **266** with TFA **274** to receive a proton provided the protonated AQ-H^+^ **268**. Light excited AQ-H^+^ **268** to the excited state AQ-H^+*^ **269**, which then oxidized arenes to give arene radical cation **271** and AQ-H^•^ radical **270**. The generated arene radical cation **271** then captured the iodine atom from iodine molecule **272** to produce iodinated product **267** and iodine radical I^•^
**273**, which recombined with another I^•^ to give I_2_ **272**. AQ-H^•^ **270** radical was oxidized by oxygen to regenerate AQ **266**.

## 7. Conclusions

A lot of breakthroughs using photocatalytic reactions have been achieved in this decade. Current developments in the application of visible light photocatalysts for halogenation, including chlorination, bromination, and iodination, are described in this study.

Visible light and photocatalyst-mediated halogenation technologies have many attractive advantages that make them good candidates to replace the old methods. Mild reaction conditions are useful in multi-step synthetic chemistry and selective halogenation at specific positions. Being able to replace hazardous or expensive chemicals is another big advantage. In addition, the excellent functional group tolerance provides the possibility of applying this protocol to various organic compounds including aliphatic C-H bonds, multiple bonds, carboxylic acids, and aromatics. In particular, it can also be applied to halogenation of natural compounds involved in various pharmaceutical applications.

The visible-light-photocatalytic halogenation technique allows the use of several halogen sources (Br_2_, Cl_2_, CBr_4_, HCl, etc.) without the need for initiators or harsh reaction conditions, thereby reducing costs and making the reactions “greener”.

On the other hand, using green solvents is also a big challenge. Organic solvents were utilized in the majority of the reactions described in this investigation. In spite of improving the efficiency of halogenation processes, most of them have negative impacts on the environment and require higher solvent removal and recovery costs. Replacing organic solvents with water would be a challenging research field in the future.

Despite considerable advancements in this procedure, there is still a lot of room for development at this point. Nearly all photocatalysts contain iridium, ruthenium, lithium, and transition metals (Cu, etc), or have a very complicated structure (Eoxin Y), resulting in high prices that obstruct their commercial implementation. Besides, the removal of catalysts from the products should be taken into account. The heterogeneous catalysts can be easily separated through filtration. Generally, homogeneous metal catalysts are typically used in the form of soluble salts or chelating complexes and can be easily eluted via extraction with water. However, the criteria for product purity are becoming increasingly stringent, particularly for medicinal items. Therefore, it would be great to come up with approaches to employ other photocatalysts, which are more popular and cost effective.

Out of the studies on visible-light-mediated photocatalytic halogenation, those on halogenation of C-H aliphatic compounds and halogenation of arenes are dominant because aliphatic and aromatic halide derivatives have a wide range of applications in medicine and industrial chemistry. In the future, there will be a need to discover additional methods for direct halogenation from other functional groups.

In conclusion, the application of visible-light-mediated photocatalysis represents a major advance in the field of halogenation of organic compounds, offering a significant difference from traditional methods. Further research in the future may help overcome the limitations of photocatalytic halogenation and expand the applications to other functional groups.

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
