# Peer review of "Visible-Light-Induced Catalytic Selective Halogenation with Photocatalyst"

_molecules, 2021, doi:10.3390/molecules26237380_

Round 1
Reviewer 1 Report
This review paper presents major advances in halogenation of a variety of organic molecules by photocatalysis. I hope some questions can be discussed in the paper before acceptance. The questions are as follow:
- What are the differences about visible light induced catalytic halogenation under reaction condition in neat water or organic solvents? And what are the advantages or disadvantages of using neat water or organic solvents?
- The iridium, ruthenium, lithium and transition metals are used during this chemical process of visible light induced catalytic halogenation. How to separate these metals from halogenated products in order for avoiding toxic side effects in the pharmaceutical industry.
- Alkenes react with CF3SO2Cl in the presence of K2HPO4 as an additive in acetonitrile. Please discuss the influence of different PH values for excited state*Ru(Phen)32+ and oxidation state Ru(Phen)33+.
- About halogenation of aromatic C-H bonds, if the Cu is changed into other transition metals, what are the impacts on yields and selectivity of halogenated products?
- The erythrosine B is used as photocatalyst with lower cost compared to the metal complexes. However, the question is whether erythrosine B is stable under long-time irradiation of light.
Author Response
(Q1) For the comment, “1. What are the differences about visible light induced catalytic halogenation under reaction condition in neat water or organic solvents? And what are the advantages or disadvantages of using neat water or organic solvents?.”,
(A1-1) The following sentences were added to explain the differences about visible light induced catalytic halogenation under reaction conditions in neat water and the advantages or disadvantages of using neat water or organic solvents on page 26 and 47.
“…Being abundant, readily obtainable, inexpensive and non-toxic, wate is more environmentally friendly compared to other organic solvents. However, most organic substrates are poorly soluble in neat water, and thus the reactions that taking place in neat water are often inefficient and generate the products with low yields. On the other hand, organic substrates are easily soluble in organic solvents such as DMF, and organic solvents help increase the reaction yields even though they are harmful and not environmentally friendly. Wu's method used water as a solvent, but could overcome this disadvantage of aqueous solvents to still achieve high yields.….”
“…On the other hand, using green solvents is also a big challenge. Organic solvents were utilized in the majority of the reactions described in this investigation. In spite of improving the efficiency of halogenation processes, most of them have negative impacts on the environment and require higher solvent removal and recovery costs. Replacing organic solvents with water would be a challenging research field in the future.….”
(Q2) For the comment, “2. The iridium, ruthenium, lithium and transition metals are used during this chemical process of visible light induced catalytic halogenation. How to separate these metals from halogenated products in order for avoiding toxic side effects in the pharmaceutical industry..”
(A2) The following sentences were added to explain the separation of metals from halogenated products about this on page 47.
“…Besides, the removal of catalysts from the products should be taken into account. The heterogeneous catalysts can be easily separated through filtration. Genetally, homogeneous metal catalysts, they are typically used in the form of soluble salts or chelating complexes and can be easily eluted via extraction with water. However, the criteria for product purity are becoming increasingly stringent, particularly for medicinal items..….”
(Q3) For the comment, “3. Alkenes react with CF3SO2Cl in the presence of K2HPO4 as an additive in acetonitrile. Please discuss the influence of different pH values for excited state*Ru(Phen)32+ and oxidation state Ru(Phen)33+..”,
(A3) We checked the reference publication (Org. Lett. 2014, 16, 1310−1313) for our review article again. However, the reference publication does not discussion or comment about the influence of different pH values for excited state*Ru(Phen)32+ and oxidation state Ru(Phen)33+. Thus, we cannot provide the discussion according to the comment from reviewer.
(Q4) For the comment, “4. About halogenation of aromatic C-H bonds, if the Cu is changed into other transition metals, what are the impacts on yields and selectivity of halogenated products?”
(A4) The following sentences were added to explain about this on page 24
“…It was reported that good yield and high regioselective halogenation of aromatic C–H bonds can be well achieved with other transition metal catalysts (Pd, Au, Ru, Co) [44-47]. However, in this study, a heterogeneous Cu-MnO catalyst was employed due to the cost-efficiency, ubiquity and versatility properties of Cu….”
(Q5) For the comment, “5. The erythrosine B is used as photocatalyst with lower cost compared to the metal complexes. However, the question is whether erythrosine B is stable under long-time irradiation of light.”,
(A5) The following sentences were added to explain about this on page 39.
“…Erythrosine B is a xanthene dye commonly used in daily life as a food colorant and a painting ink and it is difficult to degrade under visible light without other supporting agents. In this study, the optimization reactions using Erythrosine B were performed in 2, 6, and 24 hours, which gave the target products in high yields. Based on these results, it can be believed that it is stable under irradiation of light around 24 hours…”

Reviewer 2 Report
This review summarizes recent studies in visible light-induced selective halogenation (chlorination, bromination and iodination) with photocatalyst including photo-catalyzed halogenation of aliphatic C-H bonds, aliphatic multiple bonds, aromatic C-H bonds and so on. Compared with traditional protocols, the photocatalytic halogenation is a simple method under mild condition. The review summarizes the proposed mechanisms of photocatalytic halogenation and point out the research opportunities for further improving the photocatalysts performance. However, the paper is not well written. It needs much further polish before publication. And the following revisions are also need:
- I can’t image that how the structure of Ru(bpy)3Cl2 was drawn wrong in Scheme 1. That’s really a disappointing thing for the authors.
- To make it easier for readers to get information, it is recommended to add schematic diagrams of the comparison of the traditional methods with visible light-induced halogenation.
- There are lots of mistakes in the typesetting, the author should make a careful check. For example, even the words are’t line up in Scheme 57.
- Some spelling and expressions errors in the article need to be checked and fixed.
- The conclusion should be more concise and accurate.
Author Response
(Q1) For the comment, “1. I can’t image that how the structure of Ru(bpy)3Cl2 was drawn wrong in Scheme 1. That’s really a disappointing thing for the authors..”,
(A1) We revised the drawing of Ru(bpy)3Cl2 structure and insert the structure into Scheme 2. as your recommendation. (due to a change in numbering, it is now in Scheme 2).
(Q2) For the comment, “2. To make it easier for readers to get information, it is recommended to add schematic diagrams of the comparison of the traditional methods with visible light-induced halogenation..”
(A2) We added schematic diagrams of the comparison of the traditional methods with visible light-induced halogenation at Scheme 1, 15, 27, 32 and 37 in page 03, 11, 19, 22, and 25. As a result, the sequence numbers of other Schemes were also changed.
(Q3) For the comment, “3. There are lots of mistakes in the typesetting, the author should make a careful check. For example, even the words are’t line up in Scheme 57.”,
(A3) Mistakes in the typesetting were checked and corrected according to comments.
(Q4) For the comment, “4. Some spelling and expressions errors in the article need to be checked and fixed.”
(A4) Some spelling and expressions errors were checked and corrected according to comments.
(Q5) For the comment, “5. The conclusion should be more concise and accurate.”
(A5) We made conclusion section to be more concise and accurate according to comments, and the following sentences were inserted as a conclusion section rather than old conclusion.
“…A lot of breakthroughs using photocatalytic reactions have been achieved in this decade. Current developments in the application of visible light photocatalysts for halogenation including chlorination, bromination, and iodination are described in this study.
Visible light and photocatalyst-mediated halogenation technologies have many attractive advantages that make them good candidates to replace the old methods. Mild reaction conditions are useful in multi-step synthetic chemistry and selective halogenation at specific positions. Being able to replace hazardous or expensive chemicals is another big advantage. In addition, the excellent functional group tolerance provides the possibility of applying this protocol to various organic compounds including aliphatic C-H bonds, multiple bonds, carboxylic acids, and aromatics. In particular, it can also be applied to halogenation of natural compounds involved in various pharmaceutical applications.
The visible-light photocatalytic halogenation technique allows the use of several halogen sources (Br2, Cl2, CBr4, HCl and etc.) without the need for initiators or harsh reaction conditions, thereby reducing costs and making the reactions "greener".
On the other hand, using green solvents is also a big challenge. Organic solvents were utilized in the majority of the reactions described in this investigation. In spite of improving the efficiency of halogenation processes, most of them have negative impacts on the environment and require higher solvent removal and recovery costs. Replacing organic solvents with water would be a challenging research field in the future.
Despite considerable advancements in this procedure, there is still a lot of room for development at this point. Nearly all photocatalysts contain iridium, ruthenium, lithium and transition metals (Cu, etc), or have a very complicated structure (Eoxin Y), resulting in high prices that obstruct their commercial implementation. Besides, the removal of catalysts from the products should be taken into account. The heterogeneous catalysts can be easily separated through filtration. Genetally, homogeneous metal catalysts, they are typically used in the form of soluble salts or chelating complexes and can be easily eluted via extraction with water. However, the criteria for product purity are becoming increasingly stringent, particularly for medicinal items. Therefore, it would be great to come up with approaches to employ other photocatalysts, which are more popular and cost-effective.
Out of the studies on visible light mediated photocatalytic halogenation, those on halogenation of C-H aliphatic compounds and halogenation of arenes are dominant because aliphatic and aromatic halide derivatives have a wide range of applications in medicine and industrial chemistry. In the future, there will be a need to discover additional methods for direct halogenation from other functional groups.
In conclusion, the application of visible light mediated photocatalysis represents a major advance in the field of halogenation of organic compounds, offering a significant difference from traditional methods. Further researches in the future may help overcome the limitations of photocatalytic halogenation and expand the applications to other functional groups.….”

Reviewer 3 Report
Luu et. al. reporting review article the “Visible-Light-Induced Catalytic Selective Halogenation with Photocatalyst”. This review highlights the recent progress in visible light-mediated catalytic halogenation reactions. Different types of halogenation such as chlorination and bromination on aliphatic, benzylic, allylic, and aromatic C-H bond, and trifluoro chlorination dichlorination of alkene, halogenation of alcohols, carboxylic acid, and heteroaromatics have been discussed. The authors have collected most of the relevant reactions and information in the manuscript which could be useful for researchers and general readers. Hence in my opinion this review will be interested in Molecules readers and recommended for publication.
Author Response
(Q1) For the comment, “Luu et. al. reporting review article the “Visible-Light-Induced Catalytic Selective Halogenation with Photocatalyst”. This review highlights the recent progress in visible light-mediated catalytic halogenation reactions. Different types of halogenation such as chlorination and bromination on aliphatic, benzylic, allylic, and aromatic C-H bond, and trifluoro chlorination dichlorination of alkene, halogenation of alcohols, carboxylic acid, and heteroaromatics have been discussed. The authors have collected most of the relevant reactions and information in the manuscript which could be useful for researchers and general readers. Hence in my opinion this review will be interested in Molecules readers and recommended for publication..”,
(A1) We would like to thank positive comments about our manuscript.

Round 2
Reviewer 2 Report
There are some nice related work should be cited in the corresponding content, such as 1) Molecules 2019, 24, 696; doi:10.3390/molecules24040696
2) J. Org. Chem. 2021, 86, 16144−16150.
And the author has revised the manuscript which can be published after minor revision.